# The Influence of Preventive Activities on Stress Perception among Dentistry Students in the Period of the COVID-19 Pandemic

**DOI:** 10.3390/ijerph192013129

**Published:** 2022-10-12

**Authors:** Elżbieta Joanna Zarzecka-Francica, Andrzej Gala, Krzysztof Gębczyński, Małgorzata Pihut, Grażyna Wyszyńska-Pawelec

**Affiliations:** 1Department of Prosthodontics and Orthodontics, Institute of Dentistry, Jagiellonian University Medical College, 31-155 Krakow, Poland; 2Department of Conservative Dentistry with Endodontics, Institute of Dentistry, Jagiellonian University Medical College, 31-155 Krakow, Poland; 3Department of Cranio-Maxillofacial Surgery, Institute of Dentistry, Jagiellonian University Medical College, 31-155 Krakow, Poland

**Keywords:** COVID-19, pandemic, medical education, dentistry students, training, technology

## Abstract

Background: The COVID-19 pandemic resulted in the strengthening of the earlier stressors and the appearance of new pandemic-related stressors. Many students of dentistry fit the profile of a group who are particularly susceptible to stress related to the pandemic. Thus, it was necessary to implement preventive activities, reducing their stress perception. This was understood as a means of significantly influencing the student’s well-being, thus improving the quality of education. Therefore, the aim of the study was to assess the impacts of implemented preventive activities on stress perception among students of dentistry during the pandemic, as well as their influence on this assessment of the selected students’ personal experiences regarding the pandemic. Methods: The study was performed one year after the outbreak of the pandemic on students of dentistry at Jagiellonian University Medical College, Krakow, Poland. It consisted of the completion of a self-designed, voluntary, anonymous, online questionnaire. The respondent’s task was to assess the influences of implemented preventive activities on stress perception using a five-point scale. The activities were divided into external (national, global) and internal (institutional). The material was statistically analysed for all students, including pre-clinical and clinical groups. Additionally, the impact of the selected students’ personal experiences regarding the COVID-19 pandemic on the assessment was studied. Results: All preventive activities (in total) significantly reduced stress perception (*p* ≤ 0.001), but the intensity of their impacts (mean rank) was different. The most highly assessed activities were all external preventive activities, with the greatest intensity belonging to the prospect of receiving vaccination against SARS-CoV-2. The remaining external activities were the prospect of developing an effective COVID-19 treatment and the increase in knowledge about SARS-CoV-2. The internal activities were assessed and ascribed lower positions, with the exception of the theoretical classes held online. This activity was placed slightly lower than the highest-rated activity, the prospect of receiving vaccination. Conclusions: The studied preventive activities reduced stress perception with different levels of intensity. The highest-ranked activities were external activities. One exception was the theoretical classes held online, an internal activity. Due to the lower impact of the internal activities and the ability to modify them by the educators, there is a need to strengthen their effectiveness. The possibility of monitoring and tailoring some preventive activities to the students’ needs was the practical aspect of the conducted study. Students’ personal experiences related to the COVID-19 pandemic influenced their assessment of the preventive activities, in some cases significantly.

## 1. Introduction

The COVID-19 pandemic caused by SARS-CoV-2 was announced by the World Health Organization on 11 March 2020. Its outbreak caused changes at all levels of societies and among individual citizens [1,2].

New challenges arose in the areas of pre- and post-graduate medical education. Due to the mechanism of the spread of the infection, it was necessary to perform multifaceted actions in order to maintain the continuity of education, which has become a priority.

The development of mechanisms tailored to the needs of medical education took time. Therefore, in the period from March to June 2020, teaching adopted a completely online mode. In the case of the curriculum of the students of dentistry, this meant the elimination of a crucial element: practical classes, i.e., training using simulations of medical patients and clinical classes with real patients [3,4,5]. 

In the 2020/2021 academic year, the situation changed, and hybrid teaching was widely implemented in Polish medical schools. For dentistry students, this meant theoretical classes held online and pre-clinical activities using medical patient simulators, as well as clinical classes with real patients. These were situations which could have potentially enabled infection [6,7].

The approach of the students to the education process in new, biologically hazardous conditions was important. The stressors usually occurring in dental education, i.e., an awareness of the growing backlog of studies, the inability to mitigate it, and potential difficulties in passing clinical subject tests and examinations, became stronger than usual. Additionally, new pandemic stressors were identified. These were related to the return to clinical classes and contact with patients in connection with SARS-CoV-2, the fear of backlog arising from the COVID-19 pandemic, and an increased number of study-related obligations during the COVID-19 pandemic [7,8,9,10,11,12,13]. 

Despite the recognition of the large-scale, pandemic-induced stressors among the students of dentistry, there are no reports concerning the influences of preventive activities on stress perception related to the COVID-19 pandemic in this group. The lack of such reports on a new biohazardous environment is an important shortcoming on the part of medical educators and could have a significant impact on the education process. 

It is worth noting that learning in the conditions of health- and life-threatening stressors may affect the well-being of students and the process of acquiring knowledge, thus reducing the quality of education [14,15,16,17]. The situation is made worse by the fact that many dental studies students, a majority of whom are female in Poland, fit the profile of a group who are particularly susceptible to stress related to the pandemic. These include young people, women, and those living alone [18,19,20]. 

Therefore, the implementation of preventive activities reducing stress perception among this group was necessary.

The students’ feedback was crucial in order to develop an appropriate method. Therefore, we planned to investigate how students of dentistry at Jagiellonian University Medical College (JUMC) perceived the influences of implemented preventive activities on stress perception during the period of hybrid education in the COVID-19 pandemic.

It is worth noting that the monitoring of the influences and tailoring of the preventive activities to the students’ needs was the practical aspect of the conducted study.

### The Aim

The aim of the study was to assess:The impact of implemented preventive activities on stress perception among dental students during the COVID-19 pandemic;The impact of the selected students’ personal experiences regarding the COVID-19 pandemic on this assessment (student’s contraction of SARS-CoV-2 infection and disease, vaccination against COVID-19, infection of family or relatives with COVID-19, and the death of family members or relatives due to COVID-19).

## 2. Materials and Methods

The study was performed one year after the outbreak of the COVID-19 pandemic, from 22 March to 1 April 2021. It consisted of the completion of a self-designed, voluntary, anonymous, online questionnaire addressed exclusively to the students of dentistry at Jagiellonian University Medical College.

The students were informed about the study by their assistants. The link to the questionnaire was sent via e-mail to the students’ representatives for each year of the study, who forwarded it to the rest of the group. The survey was available 24 h a day, with a predefined deadline. 

Before starting the survey, the respondents were informed about the nature of the study, its objectives, and the methodology. The action of completing the questionnaire was understood as expressing informed consent to participate in it. Subjects could opt out of completing the questionnaire at any time without giving a reason. The study was approved by the Bioethics Committee of the Jagiellonian University (No. 1072.6120.120.2020).

The questionnaire consisted of two parts. In the first, there were questions about gender and the year of study and four additional questions, with the possibility of answering yes or no. The questions were as follows: Have you experienced COVID-19? Are you vaccinated against SARS-CoV-2? Has anyone from your family or any of your relatives had COVID-19? Has anyone from your family or any of your relatives passed away due to COVID-19?

The second part of the questionnaire listed eight preventive activities that enabled the return to regular education. The activities were developed based on the research we conducted in the early stages of the pandemic among students of dentistry at Jagiellonian University Medical College. This study allowed us to identify the strongest stressors [9].

The activities were divided into external (national, global) and internal (institutional).

The external activities included the prospect of vaccination against SARS-CoV-2, the prospect of developing an effective COVID-19 treatment, and the increase in knowledge about SARS-CoV-2.

The internal activities included theoretical classes held online, the return to clinical classes, act of improving the level of personal safety while performing dental procedures by implementing personal protective equipment (PPE) (aprons, masks, face shields), knowledge about appropriate equipment and sanitary products in the operation rooms (UV air filters, proper air exchange in procedure rooms, disinfectants), and training with the aim of maintaining biological safety during the pandemic within the premises of a medical facility (distancing, getting around the facility, hygiene rules).

To estimate the internal consistency of the reliability of the questionnaire, Cronbach’s α and McDonald’s ω analyses were used. Values above 0.7 are considered satisfactory for the indicators [21]. The obtained results were 0.758 for Cronbach’s α and 0.774 for McDonald’s ω, thus signifying satisfactory reliability.

In order to verify the structure of the questionnaire, an exploratory factorial analysis using oblimin rotation was carried out. KMO and Bartlett’s tests are measures of the adequacy of the selection of input variables for factor analysis. A value above 0.5 for KMO signals the validity of the factor analysis. Bartlett’s sphericity test verifies the hypothesis of a unitary correlation matrix. If it is significant, this means that the factor model is appropriate for the analysed variables. The KMO value was 0.75 and that of Bartlett’s test of sphericity was found to be χ^2^(28) = 594.98. At *p* ˂ 0.001, the result was statistically significant, confirming the validity of the factorial distinction. Based on the eigenvalues, two factors were distinguished, which explained a total of 58.14% of the variance in the stress perception results. A scree plot confirmed the two-factor structure. Both factors had satisfactory levels of reliability. The activity of theoretical classes held online was excluded from the analysis due to its low loading value (below 0.4).

The respondents’ task was to answer questions regarding the extent to which each of the presented activities affected their stress perception. For this purpose, they used a five-point scale: 0—not applicable, which could be used if the preventive activity was not relevant, 1—does not reduce stress perception, 2—slightly reduces stress perception, 3—moderately reduces stress perception, and 4—greatly reduces stress perception. 

Each of the aforementioned preventive activities were assessed separately. Neither a domain, factor, nor any psychological tool was created.

The study group comprised solely of students of dentistry at Jagiellonian University Medical College.

Two groups were created. The first included students in their first and second years of study. During this period, pre-clinical education based on the use of simulation training models dominated. In the second group were students in their third to the fifth years of study. Clinical education dominated during this period. Students performed preventive, therapeutic, and rehabilitation procedures in various fields of dentistry on real patients in clinical conditions. 

The analysis of the influences of the implemented preventive activities on the students’ stress perception was performed for the entire group (from the first to the fifth years of study), including the groups undergoing pre-clinical and clinical education. 

The sample size was calculated a priori using G*Power software version 3.1.9.4. (Düsseldorf, Germany) and specified for repeated measures analysis, assuming an effect size of f = 0.25, alpha level = 0.05, power = 0.95, and a correlation between measurements of the moderate type (0.3). The analysis showed that the minimum sample size was estimated as 120 participants for the whole group. For the comparison between the pre-clinical and clinical groups, another a priori sample size calculation was performed. This analysis showed that the minimum sample size was estimated as 208 participants for the whole group. In both cases, our sample size complied with those requirements. 

The statistical analysis of the results was performed using IBM SPSS Statistics 26.0. and Jamovi 2.0. In order to compare the preventive activities, an analysis was performed using the Friedman test combined with post hoc analysis using the Dunn test with the Bonferroni correction. The Mann–Whitney U test was used to compare the two groups undergoing pre-clinical and clinical education, as a non-parametric method that also takes into account uneven variable distributions and counts. The level of significance was α = 0.05.

## 3. Results

### 3.1. Group Characteristics

The research was carried out using a group of 233 volunteers (71.91% of all dentistry students), 191 (82.0%) of whom were women and 42 (18.0%) of whom were men. In the Polish setting, there is a gender imbalance in the field of dentistry, with a significant prevalence of women.

In the group taking part in pre-clinical education, there were 95 (40.8%) respondents, including 38 (16.3%) first-year and 57 (24.5%) second-year students. The clinical group accounted for 138 (59.2%) students among the whole sample of respondents, of which 42 (18%) were in their third, 48 (20.6%) in their fourth, and 48 (20.6%) in their fifth year of study.

Among the participants, 197 (86.3%) had been vaccinated, which was the vast majority. The reasons for some of the students’ lack of vaccination were not investigated, but it is worth noting that, as medical students, they had priority status as candidates for vaccination at that time in Poland. 

A total of 34 participants (14.6%) had contracted COVID-19, 148 (64.4%) had a family member or relative who contracted COVID-19, and 25 (10.7%) of those surveyed had a family member or relative who passed away due to COVID-19.

### 3.2. The Influences of Preventive Activities on Stress Perception among Students in All Years of Study

All preventive activities (in total) significantly reduced stress perception among students (*p* ≤ 0.001), but the intensity of their impacts (mean rank) was different.

The analysis showed significant differences between most of them. The preventive activities that were ranked the highest in terms of the reduction in stress perception were the prospect of getting vaccinated against SARS-CoV-2 and the online form of theoretical classes. The next two top-rated activities were the prospect of developing an effective COVID-19 treatment and increase in knowledge about the virus. These preventive activities reduced stress perception to a greater extent than the others. The exceptions were two activities undertaken: increasing the level of individual safety during the performance of dental procedures by implementing PPE and increasing knowledge about SARS-CoV-2 virus, where differences in the assessment of preventive activities undertaken were insignificant (*p* = 0.120).

An average level was obtained by the increase in the level of individual safety during dental procedures through the introduction of PPE, training on the maintenance of biological safety in a medical facility during a pandemic, knowledge about appropriate equipment and sanitary products in the operation rooms, and the return to clinical classes. These activities did not differ from each other in terms of their ratings by the surveyed students. The results of the analyses are presented in Table 1.

### 3.3. Pre-Clinical Education: The Influences of Preventive Activities on Stress Perception among Students in Their First and Second Years of Study

The analysis showed significant differences between most of the preventive activities undertaken. The prospect of getting vaccinated against SARS-CoV-2 and the theoretical classes held online were rated the highest. The next two highest-rated activities were the prospect of developing an effective COVID-19 treatment and the increase in knowledge about the virus. These preventive activities reduced stress perception to a greater extent than the others. 

Knowledge about appropriate equipment and sanitary products in the operation rooms was assessed lower than the theoretical classes held online (*p* = 0.041) and the prospect of getting vaccinated against SARS-CoV-2 (*p* = 0.031).

The act of improving the level of individual safety during dental procedures by introducing PPE was rated lower than the theoretical classes held online (*p* = 0.049) and the prospect of getting vaccinated against SARS-CoV-2 (*p* = 0.037).

A detailed analysis of the influences of implemented preventive activities on stress perception among first- and second-year dental students at JUMC is presented in Table 2.

### 3.4. Clinical Education: The Influences of Preventive Activities on Stress Perception among Students in Their Third, Fourth, and Fifth Years of Study

The analysis showed significant differences between most of the preventive activities undertaken. The prospect of getting vaccinated against SARS-CoV-2 and the theoretical classes held online were rated the highest. The next three highest-rated activities were the prospect of developing an effective COVID-19 treatment, the increase in knowledge about the virus, and the return to clinical classes.

An average level of effectiveness was obtained by the act of improving the level of individual safety during dental procedures by introducing PPE, training with the aim of maintaining biological safety during a pandemic in a medical facility, knowledge about appropriate equipment and sanitary products in the operation rooms, and the return to clinical classes. These preventive activities did not differ from each other in terms of the students’ ratings. The results of the analyses are presented in Table 3.

### 3.5. Comparison of Pre-Clinical and Clinical Groups

Between pre-clinical and clinical groups, the analysis showed a statistically significant difference for knowledge about appropriate equipment and sanitary products of the operation rooms. Among students in their early years of studies, this preventive activity had a greater impact on stress perception. Apart from this activity, the analysis did not reveal any significant differences between the two groups. However, students in the pre-clinical group awarded the highest score in reducing the perception of stress to increased knowledge about SARS-CoV-2, and in the clinical group to the theoretical classes held online. The lowest score in both groups was obtained by return of clinical classes. The results of the analyses are presented in Table 4.

### 3.6. The Impacts of the Selected Students’ Personal Experiences Regarding the COVID-19 Pandemic on Their Assessment of Preventive Activities

#### 3.6.1. The Impacts of Experiences Related to One’s Own SARS-CoV-2 Infection

The analysis did not reveal any significant differences between the groups. This means that, regardless of the presence or absence of the disease in the student’s personal experience, the levels of effectiveness of the preventive activities were assessed similarly.

However, among the group of students who did not experience COVID-19, the prospect of developing an effective COVID-19 treatment was ranked the highest. The knowledge about appropriate equipment and sanitary products in the operation rooms and training with the aim of maintaining biological safety during the pandemic within the premises of a medical facility were assessed only slightly lower. It is worth emphasising that the mean rank values for these activities were among the highest in our study. The theoretical classes held online received the lowest rating. In the group of people who experienced COVID-19, the situation was reversed; theoretical classes held online reduced the perception of stress most significantly, and the prospect of developing an effective COVID-19 treatment was rated the lowest. The results of the analyses are presented in Table 5.

#### 3.6.2. The Impact of Vaccination against COVID-19 

The analysis showed that vaccinated people rated the prospect of being vaccinated against COVID-19 (moderate effect) and the prospect of developing an effective COVID-19 treatment (weak effect) higher than those who did not get vaccinated. For the remaining preventive activities, the scores among both groups were similar (the differences turned out to be statistically insignificant).

However, in the group of non-vaccinated people, theoretical classes held online reduced the perception of stress most strongly, while the prospect of receiving vaccination against COVID-19 was ranked the lowest. At the same time, the mean rank of this activity was the lowest in our entire study. Conversely, in the group of vaccinated people, the possibility of vaccination was given the highest score, while the lowest rating was awarded to the return to clinical classes. The results of the analyses are presented in Table 6.

#### 3.6.3. The Impact of the Illness of Family or Relatives with COVID-19

Table 7 presents a comparison of the assessment of the implemented preventive activities by students whose relatives/family experienced COVID-19 and students whose relatives/family did not experience it.

The analysis did not reveal any significant differences between the groups. This means that, regardless of whether the relatives/family of the respondents experienced COVID-19 or not, the influences of the preventive activities on stress perception were similar.

However, it was observed that in the group of people whose family members or close relatives did not experience COVID-19, the increase in knowledge about SARS-CoV-2 had the strongest impact on the reduction in stress perception. In the group of people whose family members or loved ones experienced this disease, the theoretical classes held online constituted the most influential activity in regard to the perception of stress. In both groups, the return to clinical classes was rated the lowest.

#### 3.6.4. The Impact of the Death of a Family Member or Relative due to COVID-19

The analysis showed that people with a family member or other loved one who died as a result of COVID-19 assessed the prospect of getting vaccinated as a more effective preventive activity in reducing the perception of stress than the subgroup of people without relatives or loved ones who died from COVID-19 (weak effect). In terms of the evaluation of other activities, no differences between the groups were noted. In addition, it was observed that in the group of people with a family member or loved one who died from COVID-19, the prospect of receiving vaccination against SARS-CoV-2 reduced the perception of stress most significantly, obtaining the highest value in our entire study (mean rank 135.07). 

In this subgroup, the lowest value was recorded for the act of improving the level of personal safety while performing dental procedures by implementing PPE. Those who did not experience the death of a family member or a loved one due to COVID-19 rated theoretical classes held online the highest and the return to clinical classes the lowest. The results of the analyses are presented in Table 8. 

## 4. Discussion

Hung et al. emphasised that the COVID-19 pandemic has significantly impacted dental education [4]. The radical change in the modes of study and thus also in the students’ lives has led to complaints of a sense of loneliness, lack of motivation, restless sleep, appetite disorders, difficulties in focusing on learning, and an increase in the incidence of somatic symptoms of stress [18,19,22,23,24,25,26,27,28,29,30]. 

For educators, knowledge about this subject is important due to the impacts of these states on the processes of learning and remembering and motivation to learn, and because of their consequences on the quality of education [6,31,32]. It is also worth noting that stress can reduce the quality of medical services provided, which is important in the context of the clinical education of dental students [33].

In our study, the activity most strongly influencing students’ perception of stress was an external activity: the prospect of receiving vaccination against SARS-CoV-2. 

During the time period in which we conducted our study, the vaccination program was ongoing in Poland. It began on 27 December 2020, and the first dose was administered to a medical professional. From that moment onwards, medical staff were vaccinated [34]. In Poland, three vaccines authorised by the European Union were available: two based on an mRNA platform (BNT162b2 by BioNTech/Pfizer and mRNA-1273 by Moderna) and the adenoviral-based vector AZD1222 (ChAdOx1-nCoV by Oxford/AstraZeneca) [35,36,37]. The vaccination of the rest of the population began on 25 January 2021 with 80-year-olds, who were joined by other groups over time. These subsequent groups included 70-year-olds, younger people with chronic diseases, people who had undergone tissue and/or organ transplantation or dialysis, those who were mechanically ventilated, and oncology patients [38,39,40,41,42]. At that time, there was no freedom to choose a specific vaccine.

During our study (from 22 March to 1 April 2021), the percentage of people who were vaccinated with the first dose was 8.5% of the population, which equated to 3,213,724 individuals. Two doses had been received by 4.7% of the population, giving a total of 1769.770 people [43]. Therefore, the probability of infection was high.

In the group of surveyed students, the percentage of those vaccinated was high and amounted to 86.3%. This could have been a consequence of easy access to the vaccine, the availability of a vaccine based on an mRNA platform, the large number of women in the study group, and/or the level and type of education of the respondents.

The ease of access resulted from the vaccination campaign organised by the university, which also included students. The enabling of access to a vaccination based on an mRNA platform vaccine may have facilitated the decision to be vaccinated. In studies of the general Polish population, it was shown that BNT162b2 by BioNTech/Pfizer and mRNA-1273 by Moderna received a high level of acceptance, in contrast to AZD1222 (ChAdOx1-nCoV) by Oxford/AstraZeneca [44]. The large proportion of women and the medical nature of their studies could have also been important. Research conducted among Poles at that time showed that women, residents of large cities, people with a higher education, and healthcare workers showed a more favourable attitude towards vaccination [45]. In the context of clinical activities and the possibility of contact with asymptomatic carriers of the virus and the infectious aerosol produced during dental procedures, the awareness of the low percentage of vaccinated people in society was not without significance.

The volunteers in our study also rated the internal activity of theoretical classes held online highly. On the one hand, such a form of tuition made it possible to maintain the continuity of the theoretical part of the students’ education, while on the other hand, it reduced the probability of SARS-CoV-2 infection. In a survey of dental students by Kharma et al., 67% of volunteers preferred the use of alternative methods of learning, excluding direct contact with the patient [46]. Many studies showed that students appreciated the possibility of distance learning [4,31,32,47]. However, after two years of the pandemic, El Homossany et al. showed that, among dentistry students, 57% believed that distance learning should be continued only in regard to the theoretical part, while 1% desired continuity in regard to clinical teaching and 32% in regard both, and 10% did not want to continue remote learning [47]. In the studies of Avubduk et al., it was observed that students of dentistry preferred hybrid education [32]. Jum’ah et al. pointed out that educational videos were the most preferred form of distance education [31].

Our observations showed that the need to suddenly switch to online classes forced students to adapt quickly to new conditions, including the possession of appropriate electronic equipment or access to Wi-Fi at their place of residence. After a year of the pandemic, online education became the order of the day in many educational institutions. In the initial period, however, online education was a stressor for some students. Inadequate equipment, a lack of network connections, lack of internet access, lack of IT support from the university staff, and a low level of knowledge about the necessary IT programs were considered to be the main causes of stress [32].

Despite these problems, multicentre studies showed that 51.8% of students completed the transition to online courses without problems, and 48.3% felt that universities were well prepared for distance learning [48].

It is worth emphasising that psychological support could also be obtained remotely. In this time of social isolation, many online services and information technology solutions were adopted. Online contact with students and healthcare workers in the form of telemedicine advice, the use of programmes dedicated to patients (mHealth, or mobile health), and artificial intelligence (Tree Holes Rescue) facilitated access to medical services [25].

In conclusion, it should be emphasised that remote teaching is becoming a basic tool in the context of complete isolation and allows for the conducting of seminars, lectures, discussion panels, conferences, and even exams. Therefore, remote methods of education should be developed, especially as they are highly rated by students.

Other external activities—the increase in knowledge of SARS-CoV-2 and development of an effective COVID-19 treatment—were also high in the ranking. It can be assumed that such attitudes are the results of an evidence-based medical education, a rational assessment of the situation, and an attempt to causally solve the pandemic-related problems. This is confirmed by studies conducted by Mustafa et al. They pointed out that almost all respondents to their questionnaire (99%) agreed that education about COVID-19 is key for preventing the spread of the disease. Knowledge about the SARS-CoV-2 virus, the methods of its transmission, and infection control measures was significantly higher in the ranking among medical and dental students than among students of other medical disciplines (in descending order: medicine, dentistry, pharmacy, nursing/applied medicine). Students in their later years of study showed much more knowledge than students in their early years. Most of the study participants showed moderate and justifiable feelings about COVID-19 [49]. Concern about the ongoing provision of evidence-based research results is therefore one of the important factors influencing the perception of stress among students.

In our study, the internal activity of returning to clinical classes was rated lower than the previously discussed activities. Studies conducted in many centres have shown that the lack of clinical classes was a strong stressor for dental students [4,6,9,48,50]. As a consequence, anxiety about their future professional careers arose [7]. At this point, it is worth emphasising, once again, that the education of dental students is practical in nature. This means that clinical activities are the essence of this process, integrating theoretical knowledge with manual skills and social competence.

A temporary alternative in a situation involving difficult access to patients may be the use of classes based on medical patient simulators. These models have a body, head, and mouth with teeth mounted inside. The teeth are made of plastic. Unlike the previously used natural teeth, the artificial teeth are safer because they prevent the cross-transmission of infections. However, prefabricated teeth do not reflect the characteristics of natural tissues. Despite the large variety of phantom teeth, they sometimes lack the anatomical structures that are important for the educational process, e.g., pulp chambers [51]. 

For these reasons, it is worth noting that, as technology develops and three-dimensional (3D) printing becomes increasingly available, it is becoming possible to print individualised didactic models, including tooth models. Printable data can be obtained from radiological spatial imaging, e.g., cone beam computed tomography, used in dentistry for diagnostic purposes. Such phantom teeth can faithfully reproduce anatomical structures and pathologies, including rare pathologies. It is also possible to print using resins with different colours and degrees of hardness, which can simulate some tissue parameters [52].

Another option is the use of virtual simulators, which, due to their built-in manipulators, allow students to learn medical procedures using virtual teeth.

Medical patient simulators enable the development of clinical thinking among students and are a valuable tool for medical educators, especially during periods of hybrid teaching in crisis situations.

However, it should be noted that this method of education provides an introduction to clinical activities and can only replace such activities for a certain period of time.

It is also worth noting that the return to clinical activities allowed the students to resume their direct interpersonal relationships. Taylor et al. emphasised that “social distancing cannot mean social isolation”. It can be assumed that the student’s group is a kind of support group in a crisis situation [53].

In order to enable students to safely carry out clinical activities, it was necessary to implement methods available at that time so as to prevent the transmission of the virus in the clinical environment. In our study, these were the internal activities.

The research of Jum’ah et al. showed that dental students have knowledge about the spread of SARS-CoV-2 and consider aerosol generated during dental procedures and non-compliance with sanitation measures by some students as the main sources of infection [31]. 

In our research, the internal preventive activities, which made it possible to continue offering a clinical education, influenced stress perception less intensively. These activities consisted of increasing the level of individual safety, equipping clinical rooms with the recommended equipment and sanitary products, and continuous training in the principles of maintaining biological safety. It can be assumed that this result was due to the fact that compliance with the rules of sanitary safety is a routine requirement in medical institutions, to which students are accustomed. The reason supporting this hypothesis may be the fact that students in the pre-clinical group, who were being introduced to the principles of clinical management, perceived knowledge about appropriate equipment and sanitary products in the operation rooms as a preventive activity which reduces stress perception more intensively than students in higher years of study, who were familiar with clinical practice. It is worth noting that the specific and stricter sanitary regime in the pandemic was often associated with discomfort at work, which made it burdensome. The high percentage of vaccinated individuals among the students may have also had significance in regard to the students’ assessment of the preventive activities that increased safety during classes.

The students’ own experiences influenced the assessment of preventive activities. In our study, we decided to measure some individual variables resulting from the pandemic: personal experiences of COVID-19, vaccination against SARS-CoV-2, COVID-19 infection of family or relatives, and deaths of family or relatives due to COVID-19. However, we must be aware of the modifying influences of many other individual and environmental variables [11,18,19].

The intensity of the activities in terms of stress reduction varied between the groups. In some cases, there were statistically significant differences. The prospect of receiving vaccination against SARS-CoV-2 and the death of a family member or a loved one due to COVID-19 had prominent impacts on the assessment.

In the group of vaccinated people, the possibility of getting vaccinated was assessed significantly higher than in the group of non-vaccinated people, by whom it was ranked in the last position. It can therefore be assumed that the vaccinated people made this decision with full conviction about the need for vaccination. People who rated the possibility of being vaccinated highly also appreciated the impact of the prospect of developing an effective COVID-19 treatment in reducing the perception of stress. In unvaccinated individuals, the prospect of receiving the vaccine reduced the perception of stress the least in our entire study. 

The low ratings of these two preventive activities among the unvaccinated may have been the result of certain unanswered questions at that time. These include questions about the durability of acquired immunity, the effectiveness and safety of experimental treatments, and the reliability of studies that enabled the production and approval of vaccines in such a short space of time. The intensified activities of anti-vaccine movements at that time, as well as widespread conspiracy theories, may have also had an impact [44].

The negative experience of having a family member or a loved one pass away due to COVID-19 turned out to significantly influence the assessment of the prospect of vaccination. This preventive activity reached the highest value in our entire study among the group of people who experienced the death of a loved one. This means that it reduced the perception of stress in this group the most significantly. Death preceded by the disease, often combined with the inability to contact the patient isolated in the hospital, turned out to be persuasive proof of the irreversibility of the processes that vaccination can prevent.

People who experienced COVID-19 particularly appreciated the importance of preventive activities. In this group, there were as many as three activities that reduced the perception of stress in addition to the previously mentioned possibility of being vaccinated. These were activities that increased the safety of classes with patients.

In summary, it is worth noting that external activities (national, global) are preventive mechanisms that are, in fact, beyond the scope of our intervention. We can mainly measure and monitor their influences on stress perception. On the contrary, internal activities (institutional) are the mechanisms that educators can create and modify in terms of their scope and intensity, tailoring them to the students’ needs. The possibility of monitoring and strengthening their impacts is the practical aspect of this study.

Additionally, the knowledge obtained from this research can be applied in the planning of preventive activities in similar situations in the future. This ought to be performed independently for different environments.

The limitation of the study is the fact that, in spite of the representative group of volunteers, 28,09% of the students did not participate. 

Additionally, we used results obtained from March and April 2021, almost one year after the implementation of preventive activities in the students’ environment. The use of the results of studies performed earlier during the hybrid period of education could provide data for longitudinal observations.

There were many variables that could influence the assessment of the implemented preventive activities, e.g., mass media exposure, personal characteristic, and lifestyle. In the study, we measured some factors directly related to the pandemic.

## 5. Conclusions

The studied preventive activities reduced stress perception with different levels of intensity. The highest-ranked activities were external activities. One exception was the internal activity of theoretical classes held online.

Due to the lower impacts of the internal activities and educators’ ability to modify them, there is a need to strengthen their effectiveness. The possibility of monitoring and tailoring some preventive activities to the students’ needs was the practical aspects of the conducted study.

Students’ personal experiences related to the COVID-19 pandemic influenced their assessment of the implemented preventive activities, in some cases significantly.

## Figures and Tables

**Table 1 ijerph-19-13129-t001:** Friedman’s analysis comparing the influences of the implemented preventive activities on stress perception among students of dentistry at JUMC during the COVID-19 pandemic. Legend: *Me*—median, *IQR*—interquartile range, χ^2^—statistics of Friedman test, *p—p*-value, *W*—Kendall’s W—effect size.

	Mean Rank	*Me*	*IQR*	χ^2^	*p*	*W*
The prospect of receiving vaccination against SARS-CoV-2	6.79	4	2			
Theoretical classes held online	6.33	3	2			
The prospect of developing an effective COVID-19 treatment	5.75	3	2			
Increasing knowledge about SARS-CoV-2	5.53	3	1			
Improving the level of personal safety while performing dental procedures by implementing personal protective equipment (aprons, masks, face shields)	4.78	2	2	432.49	<0.001	0.23
Return to clinical classes	4.38	2	3			
Knowledge about appropriate equipment and sanitary products in the operation rooms (UV air filters, proper air exchange in procedure rooms, disinfectants)	4.26	2	2			
Training with the aim of maintaining biological safety during the pandemic within the premises of a medical facility (distancing, getting around the facility, hygiene rules)	3.88	2	2			

**Table 2 ijerph-19-13129-t002:** Pre-clinical education: Friedman’s analysis comparing the influences of implemented preventive activities on stress perception among students of dentistry at JUMC in their first and second years of study during the COVID-19 pandemic. Legend: *Me*—median, *IQR*—interquartile range, χ^2^—statistics of Friedman test, *p—p*-value, *W*—Kendall’s W—effect size.

	Mean Rank	*Me*	*IQR*	χ^2^	*p*	*W*
The prospect of receiving vaccination against SARS-CoV-2	6.56	4	1			
Theoretical classes held online	6.50	3	2			
The prospect of developing an effective COVID-19 treatment	6.43	3	2			
Increasing knowledge about SARS-CoV-2	5.74	3	2			
Return to clinical classes	4.91	2	1.25			
Improving the level of personal safety while performing dental procedures by implementing personal protective equipment (aprons, masks, face shields)	4.11	2	1	65.71	<0.001	0.30
Knowledge about appropriate equipment and sanitary products in the operation rooms (UV air filters, proper air exchange in procedure rooms, disinfectants)	4.07	2	1			
Training with the aim of maintaining biological safety during the pandemic within the premises of a medical facility (distancing, getting around the facility, hygiene rules)	3.54	2	1			

**Table 3 ijerph-19-13129-t003:** Clinical education: Friedman’s analysis comparing the impacts of implemented preventive activities on stress perception among students of dentistry at JUMC in their third, fourth, and fifth years of study during the COVID-19 pandemic. Legend: *Me*—median, *IQR*—interquartile range, χ^2^—statistics of Friedman test, *p—p*-value, *W*—Kendall’s W—effect size.

	Mean Rank	*Me*	*IQR*	χ^2^	*p*	*W*
The prospect of receiving vaccination against SARS-CoV-2	6.90	4	1			
Theoretical classes held online	5.99	3	2			
The prospect of developing an effective COVID-19 treatment	5.50	3	2			
Increasing knowledge about SARS-CoV-2	5.35	3	1			
Return to clinical classes	5.32	3	2			
Improving the level of personal safety while performing dental procedures by implementing personal protective equipment (aprons, masks, face shields)	5.13	3	1	227.84	<0.001	0.25
Knowledge about appropriate equipment and sanitary products in the operation rooms (UV air filters, proper air exchange in procedure rooms, disinfectants)	4.13	2	2			
Training with the aim of maintaining biological safety during the pandemic within the premises of a medical facility (distancing, getting around the facility, hygiene rules)	3.88	2	2			

**Table 4 ijerph-19-13129-t004:** U-Mann–Whitney test for comparison of implemented preventive activities on stress perception between the pre-clinical and clinical groups of students. Legend: *Me*—median, *IQR*—interquartile range, *Z*—standardized statistics of Mann–Whitney test, *p—p*-value, *r*—Glass rank-biserial correlation effect size.

	Comparison of the Groups								
	Pre-Clinical Group (*n* = 94)			Clinical Group (*n* = 138)					
	Mean Rank	*Me*	*IQR*	Mean Rank	*Me*	*IQR*	*Z*	*p*	*r*
The prospect of receiving vaccination against SARS-CoV-2	112.98	4.00	1.00	109.72	4.00	0.00	−0.41	0.683	0.03
Theoretical classes held online	112.97	3.00	2.00	116.36	3.00	2.00	−0.40	0.690	0.03
The prospect of developing an effective COVID-19 treatment	117.40	3.00	2.00	103.27	3.00	2.00	−1.70	0.090	0.12
Increasing knowledge about SARS-CoV-2	118.16	3.00	2.00	108.70	3.00	1.00	−1.11	0.266	0.07
Improving the level of personal safety while performing dental procedures by implementing Personal Protective Equipment (aprons, masks, face shields)	92.79	2.00	1.00	104.92	3.00	1.00	−1.46	0.144	0.10
Return of clinical classes	78.69	2.00	1.25	87.81	3.00	2.00	−0.99	0.320	0.08
Knowledge about appropriate equipment and sanitary products of the operation rooms (UV air filters, proper air exchange in procedure rooms, disinfectants)	116.19	2.00	1.00	97.50	2.00	2.00	−2.25	**0.024**	0.16
Training within the scope of maintaining biological safety during the pandemic within the premises of a medical facility (distancing, getting around the facility, hygiene rules)	111.51	2.00	1.00	102.16	2.00	2.00	−1.13	0.258	0.08

**Table 5 ijerph-19-13129-t005:** Mann–Whitney U test for the comparison of the impacts of the implemented preventive activities on stress perception between volunteers who experienced COVID-19 and students who did not experience COVID-19. Legend: *Me*—median, *IQR*—interquartile range, *Z*—standardized statistics of Mann–Whitney test, *p—p*-value, *r*—Glass rank-biserial correlation effect size.

	COVID-19 Experience								
	No (*n* = 199)			Yes (*n* = 34)					
	Mean Rank	*Me*	*IQR*	Mean Rank	*Me*	*IQR*	*Z*	*p*	*r*
The prospect of receiving vaccination against SARS-CoV-2	118.01	4.00	2.00	111.12	3.00	1.25	−0.60	0.550	0.04
Theoretical classes held online	114.51	3.00	2.00	131.59	2.50	1.25	−1.43	0.153	0.09
The prospect of developing an effective COVID-19 treatment	118.75	3.00	2.00	106.75	2.50	1.25	−0.99	0.321	0.06
Increasing knowledge about SARS-CoV-2	118.28	3.00	1.00	109.50	2.00	1.00	−0.73	0.467	0.05
Improving the level of personal safety while performing dental procedures by implementing personal protective equipment (aprons, masks, face shields)	117.65	2.00	2.00	113.19	2.00	2.00	−0.37	0.712	0.02
Return to clinical classes	118.27	2.00	3.00	109.57	2.00	3.00	−0.71	0.477	0.05
Knowledge about appropriate equipment and sanitary products in the operation rooms (UV air filters, proper air exchange in procedure rooms, disinfectants)	114.71	2.00	2.00	130.38	2.00	2.00	−1.30	0.193	0.09
Training with the aim of maintaining biological safety during the pandemic within the premises of a medical facility (distancing, getting around the facility, hygiene rules)	114.85	2.00	2.00	129.59	2.00	2.00	−1.23	0.219	0.08

**Table 6 ijerph-19-13129-t006:** Mann–Whitney U test for the comparison of the implemented preventive activities on stress perception between volunteers who were vaccinated or were not vaccinated against COVID-19. Legend: *Me*—median, *IQR*—interquartile range, *Z*—standardized statistics of Mann–Whitney test, *p—p*-value, *r*—Glass rank-biserial correlation effect size.

	Vaccinated against COVID-19								
	No (*n* = 32)			Yes (*n* = 197)					
	Mean Rank	*Me*	*IQR*	Mean Rank	*Me*	*IQR*	*Z*	*p*	*r*
The prospect of receiving vaccination against SARS-CoV-2	50.67	2.00	2.00	119.04	4.00	1.00	−5.65	**<0.001**	0.38
Theoretical classes held online	112.59	3.00	1.75	115.39	3.00	2.00	−0.23	0.816	0.02
The prospect of developing an effective COVID-19 treatment	75.27	2.00	1.00	114.00	3.00	2.00	−3.18	**0.001**	0.22
Increasing knowledge about SARS-CoV-2	109.09	2.50	1.00	112.99	3.00	1.00	−0.31	0.757	0.02
Improving the level of personal safety while performing dental procedures by implementing personal protective equipment (aprons, masks, face shields)	93.63	2.00	1.00	102.14	3.00	1.00	−0.75	0.455	0.05
Return to clinical classes	85.76	2.00	2.00	86.03	3.00	1.00	−0.02	0.981	0.00
Knowledge about appropriate equipment and sanitary products in the operation rooms (UV air filters, proper air exchange in procedure rooms, disinfectants)	99.18	2.00	1.50	104.66	2.00	1.00	−0.45	0.651	0.03
Training with the aim of maintaining biological safety during the pandemic within the premises of a medical facility (distancing, getting around the facility, hygiene rules)	103.06	2.00	2.00	105.86	2.00	2.00	−0.24	0.813	0.02

**Table 7 ijerph-19-13129-t007:** Mann–Whitney U test for the comparison of the implemented preventive activities on stress perception between volunteers whose family or relatives had/has not experienced COVID-19. Legend: *Me*—median, *IQR*—interquartile range, *Z*—standardized statistics of Mann–Whitney test, *p—p*-value, *r*—Glass rank-biserial correlation effect size.

	COVID-19 Experience of Relatives or Family Members								
	No (*n* = 81)			Yes (*n* = 148)					
	Mean Rank	*Me*	*IQR*	Mean Rank	*Me*	*IQR*	*Z*	*p*	*r*
The prospect of receiving vaccination against SARS-CoV-2	107.28	3.50	1.00	113.03	4.00	1.00	−0.71	0.481	0.05
Theoretical classes held online	112.28	3.00	2.00	116.49	3.00	2.00	−0.48	0.630	0.03
The prospect of developing an effective COVID-19 treatment	116.62	3.00	2.00	104.81	3.00	2.00	−1.38	0.167	0.09
Increasing knowledge about SARS-CoV-2	120.41	3.00	2.00	108.27	3.00	1.00	−1.39	0.165	0.09
Improving the level of personal safety while performing dental procedures by implementing personal protective equipment (aprons, masks, face shields)	95.03	2.00	1.00	114.19	2.00	1.00	−1.12	0.261	0.08
Return to clinical classes	86.54	3.00	3.00	85.68	3.00	1.00	−0.11	0.909	0.01
Knowledge about appropriate equipment and sanitary products in the operation rooms (UV air filters, proper air exchange in procedure rooms, disinfectants)	103.15	2.00	2.00	104.42	2.00	1.00	−0.15	0.879	0.01
Training with the aim of maintaining biological safety during the pandemic within the premises of a medical facility (distancing, getting around the facility, hygiene rules)	108.80	2.00	2.00	103.70	2.00	1.75	−0.62	0.538	0.04

**Table 8 ijerph-19-13129-t008:** Mann–Whitney U test for the comparison of the implemented preventive activities on stress perception between volunteers with a family member or relative who died from COVID-19 and those without. Legend: *Me*—median, *IQR*—interquartile range, *Z*—standardized statistics of Mann–Whitney test, *p—p*-value, *r*—Glass rank-biserial correlation effect size.

	Death of Relative or Family Member COVID-19								
	No (*n* = 204)			Yes (*n* = 25)					
	Mean Rank	*Me*	*IQR*	Mean Rank	*Me*	*IQR*	*Z*	*p*	*r*
The prospect of receiving vaccination against SARS-CoV-2	108.20	4.00	1.00	135.07	4.00	0.00	−2.10	0.035	0.09
Theoretical classes held online	113.12	3.00	2.00	130.38	3.00	1.50	−1.29	0.197	0.09
The prospect of developing an effective COVID-19 treatment	107.44	3.00	2.00	120.98	4.00	2.00	−1.06	0.290	0.07
Increasing knowledge about SARS-CoV-2	111.81	3.00	1.00	118.00	3.00	2.00	−0.47	0.640	0.03
Improving the level of personal safety while performing dental procedures by implementing personal protective equipment (aprons, masks, face shields)	102.37	3.00	1.00	89.84	2.00	1.00	−1.01	0.314	0.07
Return to clinical classes	86.54	1.00	1.00	90.07	3.00	1.50	−0.42	0.677	0.03
Knowledge about appropriate equipment and sanitary products in the operation rooms (UV air filters, proper air exchange in procedure rooms, disinfectants)	103.92	2.00	1.00	104.61	2.00	1.00	−0.05	0.957	0.00
Training with the aim of maintaining biological safety during the pandemic within the premises of a medical facility (distancing, getting around the facility, hygiene rules)	106.65	2.00	2.00	95.66	2.00	0.25	−0.85	0.395	0.006

## Data Availability

This study has no additional supporting data.

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
