# Peer review of "The Influence of Preventive Activities on Stress Perception among Dentistry Students in the Period of the COVID-19 Pandemic"

_ijerph, 2022, doi:10.3390/ijerph192013129_

Round 1

Reviewer 1 Report

The authors present an interesting approach to a health issue and assess the impact of the preventive activities (implemented during educational process) on stress perception among dental students.

Regarding the manuscript there are some ideas the authors could clarify and strengthen in order to have a more solid manuscript:

-       The instrument was conformed by 14 questions, which was design ad hoc this study. How did the authors select and determined the included topics? Did they perform a pilot, a expert judgment analysis or any validation a priori?  Without this information there is no certainty of the reliability of the provided data and therefore the investigation.

-       In this sense the authors propose as an outcome variable the stress perception on the students, how do they differ it from “risk perception”, “anxiety” or “risk awareness”, which have been assess on several studies.  The construction of the concept is important to ensure that what is proposed is measured. How do they ensure that all the participants reported “stress perception” and avoid misconceptions?

-       It is noted that authors classified preventive actions into two aspects: external and internal, however, there are some variables that could influence on these and that has been assessed and associated such as: mass media exposure, personal characteristics, health status or COVID-19 knowledge, those with several conditions, wellbeing or lifestyle or even personality. These factors could affect the weight that the activities have will have a lower impact on the perception of stress.

-       On the other hand, it would be useful to present a multivariate analyzes that allow evaluating together how these activities impacted on the stress of the students, all together or into categories (internal and external) as they classified it.

-       Beyond the evaluation, the authors could include a section to state what it would be the possible application of the presented results, to emphasize the need and usefulness for this information.

Author Response

Thank you for all your questions and suggestions regarding our study. They will help us improve the quality and value of our manuscript.

1. Thank you for your question regarding the design of our questionnaire. The questions in it were developed based on research we conducted in the early stages of the pandemic among students of dentistry at Jagiellonian University Medical College. This research allowed us to identify the strongest stressors. For this purpose, we made use of a validated tool , the Dental Environmental Stress Questionnaire  (DES) and 11 supplementary questions related to the COVID-19 pandemic (Dental Environmental Stress during the COVID-19 pandemic at the Jagiellonian University Medical College, Kraków, Poland published in the International Journal of Occupational Medicine and Environmental Health doi: 10.13075/ijomeh.1896.01773.). In this study the domain Academic work presented the highest level of stressors. The fear of backlog, the fear of failing a course or the entire year, and examinations were the strongest of these.

Apart from identifying the strongest stressors using the DES, we also studied the correlation of these stressors with additional questions related to the pandemic. The strongest positive relationships between the supplementary questions and the DES domains appeared in the area of clinical factors. These concerned returning to clinical classes and contact with patients with regard to SARS-CoV-2 (rs = 0.50, p˂0.001), contact with patients in connection with SARS-CoV-2 (rs=0.47, p˂0.001), and a lack of practical classes with patients in connection with the COVID-19 pandemic (rs= 0.42, p˂0.001).

The results of a previous study we had conducted allowed us to define preventive activities which may have impacted a reduction in the perception of stress related to the effects of the stressors we identified in our previous study.

As active medical educators with many years of experience, we had at our disposal knowledge about the possibilities and manner in which these activities could be implemented in an educational environment.  

In summary, we came to a consensus about the form of the questions by analysing: stressors in the times of the pandemic, the curriculum of students of dentistry, the clinical conditions in which the students’ education was conducted, and the principles of the pandemic hygiene regimen at medical facilities. (lines 116-119)

To estimate the internal consistency of reliability of the questionnaire, the Cronbach’s α and McDonald’s ω analysis were used. The obtained results were 0.758 for  Cronbach’s α and 0.774 for McDonald’s ω, and signify satisfactory reliability. (lines 132-134)

In order to verify the structure of the questionnaire, an exploratory factorial analysis with Oblimin rotation  was carried out. The KMO value was 0.75 and Bartlett’s test of sphericity was found to be χ²(28)=594.98; pË‚0.001 statistically significant, confirming the validity of the factorial distinction. Based on the eigenvalues, two factors were distinguished which explained a total of 58.14% of the variance of the stress perception. The scree plot confirmed the two-factor structure. Both factors had satisfactory levels of reliability. The activity ‘theoretical classes held online’ was excluded from the analysis due to it’s low loading value (below 0.4) (in the attachments, Fig.1).(lines135-142)                   The reliability analysis for the Factor 1 was as follows: Cronbach’s α= 0.751, McDonald’s ω= 0.791 and for the Factor 2: Cronbach’s α= 0.785, McDonald’s ω = 0.797.

We also calculated the sample size appropriate for the research. It was calculated a priori using G*Power software version 3.1.9. and specified for repeated measures analysis assuming an effect size of: f=0.25; alpha level = 0.05; power = 0.95 and a correlation between measurements of the moderate type (0.3). The analysis showed that the minimum sample size was estimated as 120 participants for the whole group. For the comparison between the pre-clinical and clinical groups, another a ’priori sample size calculation was performed. This analysis showed that the minimum sample size was estimated as 208 participants for the whole group. In both cases, our sample size complies with those requirements. (lines 161-168)

2.  Regarding doubts about an unambiguous understanding of the subject of the study, we reply that the text of the questionnaire unambiguously defines what will be studied, in what way, and using what measures. (lines 143-147)

It is worth noting here that our aim was a measurement of the perception of stress, and not of the degree of stress. In the Polish setting, the measurement of the degree of stress is the province of psychologists. There are tests for this purpose published by the Psychological Test Laboratory of the Polish Psychological Association. (lines 148-149)

For us, as medical educators, the important aspect was the attitude of students and their feelings in the context of the applied preventive activities. The positive assessment of these measures signalled their effectiveness, in turn indicating a potential positive impact on the students’ well being. (lines 87-92)

3.Thank you for bringing our attention to the fact that many variables can influence personal stress perception. The action of these variables is very diverse, and personal susceptibility to them is individual. While our study allowed for appropriate modification of the intensity of some preventive activities with regard to the needs of the studied groups, nonetheless it seems that individual support should be left to specialists in psychology.  It is well-known, after all, that the individual perception of stress is shaped by a wide range of personal variables.  (lines 482-484, 528-530, 148-149)

Nevertheless, the results of our study should be interpreted with reference to the entire study group and to the isolated subgroups. From the point of view of educators, the relevant aspects for us were those which had a direct impact on the realisation of the students’ program of study, and which can be modified as necessary. We were interested in the practical application of the results obtained. (lines 517-518)

 4 and 5. Thank you very much for your comment regarding the need for greater emphasis on the distinction between activities into external and internal, and the possible application of the presented results. This led us to a new way of understanding and interpreting the results we obtained and of formulating new conclusions. 

We noticed that, external activities (national, global) comprised preventive mechanisms which were independent of us. We could mainly measure and monitor their influence on stress perception. Internal activities (institutional) comprised mechanisms which we could create ourselves, whose scope and intensity could be modified and tailored to the students’ needs. (lines 513-518)

The possibility of monitoring the influence of preventive activities on stress perception and of strengthening their impact is the practical aspect of the conducted study. The almost all internal activities were assessed relatively low. This may indicate a need for medical educators to strengthen them.

In our opinion, research on the net impact of external and internal activities will allow us to obtain results without assessment of individual activities. This will prevent us from indicating specific areas which should be strengthened.

We hope that we have followed the suggestions and answered the questions with satisfactory way.

With kindest regards

Elżbieta Zarzecka-Francica

Reviewer 2 Report

Comments to the Author

Thank you for providing me with the opportunity to evaluate this document. Even though the topic is essential, the manuscript has significant flaws that limit the study's value and contribution. Please see my thorough remarks for suggestions on how to improve. Kindly explain - The exclusion criteria used for sampling. The introduction is not for you to explain these general knowledge terms but to explain why the topic of your research is essential, what gaps are there, and why your research can fill these gaps. This way of writing may not meet the quality requirements of the journal. Why is it that investigating can make a significant contribution to the literature on consumer behavior? However, you can find much relevant research for every variable of your theoretical framework, which leads to too low incremental contribution to your theoretical framework. Your theory development is weak. Because there is no single theory to support your whole theoretical model, the data should be a longitudinal section to confirm the causal relationship in your theoretical model. Why did you not mention Common Method Variance? Discussion 1. You only intruded on the analysis results using these models, but you should further explain why these results are so crucial that they can open the black box of consumer behavior. The English are rough, and the paper needs a thorough revision to address that.

Many statements need to be argued thoroughly. If not, it is one clear example of the need for careful editing of the English, preferably by a native English speaker.!! As it stands, most of the findings back up earlier research. I suggest you should rewrite the introduction to demonstrate the contributions. The paper's total contribution is limited to confirming earlier findings. What different results would one expect with a sample from a developed country? What steps did the authors take to ensure the sample is representative of the population? The research gap addressed by this study should be recognized more prominently. The results are predictable. The discussion reveals that most of the findings confirm, are consistent with, or are in line with earlier studies.

Title:

- The title does not reflect the study's aim and scope, so the title should be changed.

Abstract:

- The abstract is not well structured.

Introduction:

- There are many unnecessary discussions.

- The motivation of the study is not well articulated, and why Iran as the study sample is missing.

- The contribution of the study is missing; add contributions of the study after drawing results in the introduction. Overall, authors may follow the outline to modify the introduction:

(i) a summary of the theoretical controversy on the issue (which you will finally say remains unresolved)

(ii) a summary of the empirical controversy on the issue (which you will finally say remains unresolved)

(iii) The significant questions or issues (derived from the above theoretical and empirical controversies) that you want to address

(iv) How is your work different from, an extension to, or an improvement on others' works (i.e., your contributions to the literature)? Here try to put in the literature
gaps you are trying to fill (including applications of any current method).

(v) a summary of the significant findings of your paper

(vi) the structure of your paper

Literature Review:

- The literature review is inferior. Authors directly moved without acknowledging theory and conventional models. Discuss related theory and literature only. 

Author Response

Thank you for your thorough and correct remarks for suggestions concerning improving of our manuscript. This led us to a new way of understanding and interpreting the results we obtained and of formulating new conclusions. 

We have introduced the following changes:

Title – the title has been changed and we hope that in its new form it will more precisely characterise the presented study. (lines 2-4)

Abstract – the abstract has been completely rewritten and presented in a structural manner. (lines 14-47)

Introduction – the Introduction has been modified to demonstrate the contribution of the study to the literature.  The narrative is as follows:

  1. emphasis on a crucial problem in the education of dentistry students during total lockdown and remote learning, (lines 57-60)
  2. emphasis on a crucial problem in the education of dentistry students during hybrid learning, (lines 61-65)
  3. focus on the strengthening of “old” and appearance of “new” pandemic-related stressors in dentistry students’ academic environment, (lines 66-73)
  4. the need to reduce the intensity of the effect of the stressors, (lines 74-81)
  5. the indication of the group of students of dentistry as one which is particularly susceptible to the impact of stressors during the COVID-19 pandemic, (lines 81-84)
  6. the emphasis on the need to gather the opinions of students in order to assess the effectiveness of activities aimed at reducing the perception of stress (the feedback), (line 87)
  7. focus on the fact that the obtained results allow for the modification of specific preventive activities to the needs of the students. In this way, the study acquires a practical value. In the absence of similar reports we point out its innovative character. (lines 91-92)

We hope that in this form we will clearly justify the need for conducting our study.

Materials and Methods:

Exclusion criteria – our study was addressed exclusively to dentistry students.  The construction of the program in which the questionnaire was presented did not allow students to continue to a subsequent question without answering the current question. In this situation, it was also not possible to submit the questionnaire. Analysis was performed on all questionnaires submitted.(lines 150-151)

We would like to say that our research method is repeatable and therefore we can plan further studies in the future which will be the basis for a longitudinal study. In the article submitted for review, we make use of the results from March and April 2021 and assess the influence of implemented preventive activities on stress perception at that time, so the Common Variance Method is not applicable in our research. The idea for using our results to continue the study is highly valid and we thank you for it. We also pointed out in the limitations of the study that we analise the sample from March and April 2021. The results of the study performed earlier during hybrid period of education could give the data for longitudinal observations. (lines 524-527)

We  calculated the sample size appropriate for the research. It was calculated a priori using G*Power software version 3.1.9. and specified for repeated measures analysis assuming an effect size of: f=0.25; alpha level = 0.05; power = 0.95 and a correlation between measurements of the moderate type (0.3). For the comparison between the pre-clinical and clinical groups, another a ’priori sample size calculation was performed. This analysis showed that the minimum sample size was estimated as 208 participants for the whole group. In both cases, our sample size complies with those requirements. (lines 161-168)

Discussion

We have included in the discussion the significant findings of our research and their innovative character, and we highlighted the possibility of using them in practice, clearer differentiation activities into external and internal, which influenced the modification and interpretation of the results and conclusions.  This enabled us to make a significant contribution to the literature. (lines 513-521)

Literature

We have completed the literature, using more relevant research.

 We have also asked a native English speaker to revise our manuscript in language terms.

We hope that we have followed the suggestions and answered the questions with satisfactory way.

With kindest regards

Elżbieta Zarzecka-Francica

Round 2

Reviewer 2 Report

Good job

Author Response

Thank you, please check the attachment.
